Energy-efficient routing protocol for reliable low-latency Internet of Things in oil and gas pipeline monitoring

Karam Sana Nasim 1 2
Bilal Kashif 2
Khan Abdul Nasir 2
Shuja Junaid junaid.shuja@utp.edu.my 3
Abdulkadir Said Jadid 3
1 Department of Computer Science, Allama Iqbal Open University , Islamabad , Pakistan
2 Department of Computer Science, COMSATS University Islamabad, Abbottabad Campus , Abbottabad , Pakistan
3 Department of Computer and Information Sciences, Universiti Teknologi PETRONAS , Seri Iskandar , Malaysia
Asif Muhammad
Electronic publication date: 2024 Feb 29
Publication date: 2024
Volume: 10
Electronic Location ID: e1908
Received 2023 Nov 13; Accepted 2024 Jan 31
Copyright: ©2024 Karam et al.
Copyright year: 2024
Copyright holder: Karam et al.
License: This is an open access article distributed under the terms of the Creative Commons Attribution License, which permits unrestricted use, distribution, reproduction and adaptation in any medium and for any purpose provided that it is properly attributed. For attribution, the original author(s), title, publication source (PeerJ Computer Science) and either DOI or URL of the article must be cited.
License URL: https://creativecommons.org/licenses/by/4.0/

Keywords: Routing, Internet of Things

Funding: The authors received no funding for this work.

==============================
The oil and gas industries (OGI) are the primary global energy source, with pipelines as vital components for OGI transportation. However, pipeline leaks pose significant risks, including fires, injuries, environmental harm, and property damage. Therefore, maintaining an effective pipeline maintenance system is critical for ensuring a safe and sustainable energy supply. The Internet of Things (IoT) has emerged as a cutting-edge technology for efficient OGI pipeline leak detection. However, deploying IoT in OGI monitoring faces significant challenges due to hazardous environments and limited communication infrastructure. Energy efficiency and fault tolerance, typical IoT concerns, gain heightened importance in the OGI context. In OGI monitoring, IoT devices are linearly deployed with no alternative communication mechanism available along OGI pipelines. Thus, the absence of both communication routes can disrupt crucial data transmission. Therefore, ensuring energy-efficient and fault-tolerant communication for OGI data is paramount. Critical data needs to reach the control center on time for faster actions to avoid loss. Low latency communication for critical data is another challenge of the OGI monitoring environment. Moreover, IoT devices gather a plethora of OGI parameter data including redundant values that hold no relevance for transmission to the control center. Thus, optimizing data transmission is essential to conserve energy in OGI monitoring. This article presents the Priority-Based, Energy-Efficient, and Optimal Data Routing Protocol (PO-IMRP) to tackle these challenges. The energy model and congestion control mechanism optimize data packets for an energy-efficient and congestion-free network. In PO-IMRP, nodes are aware of their energy status and communicate node’s depletion status timely for network robustness. Priority-based routing selects low-latency routes for critical data to avoid OGI losses. Comparative analysis against linear LEACH highlights PO-IMRP’s superior performance in terms of total packet transmission by completing fewer rounds with more packet’s transmissions, attributed to the packet optimization technique implemented at each hop, which helps mitigate network congestion. MATLAB simulations affirm the effectiveness of the protocol in terms of energy efficiency, fault-tolerance, and low latency communication.

Introduction

Oil and gas industry (OGI) pipelines are the primary resource for global energy transportation (Ravishankar et al., 2022). Currently there are 2.2 million miles of pipelines, deployed to fulfill the continuous demand for OGI products around the world (Mohamed El Amine Ben Seghier, Daniel Höche, 2022). Pipelines are considered the most economical, faster, and easiest way of transporting oil and gas from production facilities to the consumers than trucks and trains. They bear the responsibility of transporting approximately 64% of energy commodities in the United States (Ravishankar et al., 2022). However, pipelines made of stainless steel are vulnerable to leakages. Pipeline leakages can be caused due to several reasons such as corrosion, material weld failure, excavation damage, natural force damage, equipment failure, and due to bad installation. Detecting these potential issues is crucial, temperature and pressure are fundamental parameters in OGI that play a pivotal role in identifying leakages. Fluctuations in temperature and pressure values at specific points within the pipeline indicate potential leakage events. Pipeline leakage is indeed a serious event and has its impacts on humans, infrastructure, wildlife, and vegetation (Yang & Suzhen Li, 2022). As an example, only in the USA, 745 pipeline leakage incidents occurred between 1994 and 2013 resulting in around 1059 injuries, and a monetary loss of about 110 million USD (Al-Sabaeei et al., 2023). Therefore, pipelines require rigorous preventive measures, monitoring, and emergency response planning to minimize the associated risks to societies, the environment, OGI assets, and workers.

Accuracy in determining the leakage points and fast data transmission are the two important aspects of pipeline monitoring for swift actions to avoid loss. Depending on human operators to conduct OGI pipeline parameter inspections to achieve accuracy becomes a challenging, costly, and slow endeavor (Al-Sabaeei et al., 2023; Minhas & Naveed, 2023). Autonomous data collection and data processing are not new to OGI. A notable example is the longstanding use of supervisory control and data acquisition (SCADA) systems for asset monitoring (Wosowei & Shastry, 2023). Moreover, wireless sensor network (WSN) based monitoring and data collection approaches have also been considered for pipelines (Oyubu et al., 2022). Both methods offer data accuracy; nevertheless, achieving swift data communication remains a challenge in WSNs compared to SCADA. However, deploying SCADA systems and conventional WSNs comes with its own set of challenges including incompatibility, high equipment and maintenance costs, scalability, non-uniformity, lack of synchronized communication, and operational coherence within their operational domains and across processes in WSNS (Wadhaj, Thomson & Ghaleb, 2022).

The Internet of Things (IoT) has become a hot research area after the fourth industrial revolution, also known as “Industry 4.0”, has been applied in various fields of life and OGI pipeline inspection is one of these fields (Chehri et al., 2021). The emergence of IoT in OGI offers a solution to the limitations connected with SCADA and conventional WSN systems in terms of cost-effectiveness, interoperability, synchronized connections, and operational coherence. IoTs have the ability for self-organization and network reconstruction in OGI pipeline dynamic scenarios. They can be installed on transmission pipelines in challenging environments and deployed from remote locations, these devices enable continuous monitoring, providing precise data for detecting pipeline leaks (Khan et al., 2017). However, IoTs installed along extensive pipelines that traverse hazardous environments entail their own range of challenges.

IoT devices are restricted by limited battery life and are prone to failures, which requires regular human intervention for maintenance (Kreis et al., 2021; Shamisa & Aliakbar, 2017). Energy efficiency and fault tolerance are the common challenges of IoT-based applications, and many studies have focused on them (Ahmed et al., 2022; Vaibhav, Shashikala & Prasenjit , 2022). However, OGI-specific environmental conditions significantly change the meaning of these challenges. In this research, we address challenges of energy efficiency, fault tolerance, low latency, and optimal data transmission specific to challenging configurations of the OGI pipeline (Balakrishnan et al., 2023; Ali & Ali, 2023).

IoT plays a crucial role in monitoring lengthy OGI pipelines situated in hazardous environments. The linear configuration of these devices means that there is no overlap in their monitoring regions. The depletion of energy from a single IoT node can have a significant impact on the overall monitoring operation. Therefore, the key challenge in this context is to ensure energy-efficient and fault-tolerant communication, given the difficulty of managing and restoring defective IoT devices scattered across the extensive pipeline, either through recharging or replacing batteries, as needed (Hu et al., 2023). Furthermore, the data collected by these IoT devices can be categorized as normal or critical. Considering that OGI pipelines pass through multiple signal blind areas, it becomes crucial to ensure the timely transmission of critical data to alleviate latency issues. Additionally, addressing the challenge of optimal data transmission is essential in OGI environments. The continuous sensing and transmission of data by IoT devices may lead to the repeated sensing of redundant data, resulting in unnecessary energy consumption and network congestion if not effectively managed. Therefore, in this article, a priority-based, energy-efficient, and optimal data routing protocol is presented for a robust IoT network for linear OGI pipelines, where routing is determined by the content type. In the proposed PO-IMRP protocol, energy efficiency is obtained by introducing packet optimization and intelligent data transmission of IoT nodes. In addition, we attempt to inform about node depletion status beforehand for a robust IoT network. We use a priority-based routing technique to determine low-latency routes for critical data. Optimal data transmission is done by removing redundant data and by maintaining a time-lapse for normal data acquisition and transmission thereby reducing traffic in the main network. Therefore, a considerable reduction in energy consumption, latency, and network traffic could be obtained besides providing a robust IoTs network for OGI. The main contributions of this article are summarized as follows:

• Proposing a packet optimization and priority-based, energy-efficient IoTs routing model for OGI pipelines surveillance system that can reduce traffic, delay and IoTs energy consumption.

• Intelligent route selection technique for critical data to provide low latency communication mechanism.

• Increasing network efficiency in terms of optimal data transmissions of packets in the IoT network.

The remainder of this article is organized as follows. In ‘Related work’, literature related to the proposed method is reviewed in terms of its limitations and advantages. ‘IoTs System Design for OGI Pipelines’ presents the proposed IoTs system design for OGI pipelines. In ‘Simulations & Results’, we conduct the implementation of the proposed methods, and its evaluations are carried out. Finally, the article concludes in ‘Conclusion’.

Related Work

Aba et al. (2021) used pressure pulses based on the notion of pipe vibration. Sensor networks are utilized to locate bursts, leaks, and other anomalies (damages) in typical pipeline systems. In this work, the principle of the temporal delay between pulse arrivals at sensor sites was used. The capacity to execute real-time damage location using a combination of wave propagation, an active sensor network, a wireless data transmission system, and an IoT platform is utilized. The transmission of data from sensors to the monitoring room is done remotely, enabling real-time monitoring of pipelines around the world. The WiFi module is activated by an internet connection from an Android phone. However, limited internet availability along OGI pipelines is one of its limitations. Additionally, there are no mechanisms for energy-efficient and fault-tolerant IoT communication. The ThingSpeak IoT analytics platform used to make decisions in this work offers high latency as well.

Spandonidis et al. (2022) designed an energy-efficient wireless sensor system that is deployed in a noisy industrial environment to enable immediate leak detection in metallic piping systems used for transporting liquid and gaseous petroleum products. Two distinct artificial intelligence (AI) models are used to detect leakage. Sensor communication along long pipelines in diverse environments is a challenging task. However, no communication mechanism is presented for energy efficiency, fault tolerance, and low latency communication. An Integrated IoT-based intelligent architecture to perform online monitoring and control of pressure and flow rate in the fluid transportation system is developed in the study Priyanka et al. (2020). The IoT-based architecture incorporates SCADA with an LQR-PID controller, serving as the local control unit or local intelligence. Architecture is monitored through a high-level online server IoT interface to detect pipeline leaks and cracks early, preventing catastrophic situations caused by drastic pressure and flow rate changes. A smart IoT module is also developed to enhance data communication between the cloud server and the pipeline hardware setup. Upon the detection of cracks or leaks at the IoT front end, the cloud server promptly activates an emergency shut-off mechanism through the smart IoT module, effectively halting the pump operation. However, energy efficiency, system robustness, and low latency are not considered in this research.

Khan et al. (2017) present an IoT-based monitoring system that includes smart objects, gateway modules, and control centers. The applications running on the smart object and gateway modules will perform real-time actions (fire alarms, equipment shutdowns, staff evacuations, and fault localization) in response to abnormal occurrences such as OGI leaks and fire hazards. However, a key challenge is that smart objects are energy-constrained, and no efficient method for energy-efficient data transfer has been implemented. Singh et al. (2021) introduced the 2.4 GHz-based Zigbee and 433 MHz-based LoRa that are employed for communication, resulting in a cloud-enabled hybrid architecture. Zigbee is a low-power wireless communication technology based on IEEE 802.15.4 that establishes a personal area network (PAN). However, no energy-efficient and fault-tolerance mechanism is presented.

IoT solutions have emerged as a cost-effective and efficient method for data gathering. However, OGI pipelines often span vast distances, including remote and communication-limited areas like deserts where IoTs are particularly vulnerable to end-to-end delays and congestion because they possess a limited number of paths leading to the sink. This fact drastically changes the problem definition of such IoT network in terms of energy efficiency, fault tolerance, and low latency (Shuja et al., 2021). Numerous energy-efficient and resilient IoT routing protocols have been presented in the literature, such as the Energy Efficient Double Rounds Clustering Protocol (EEDRCP) (Chen, Zhao & Chen, 2010) and the pre-partition-based uneven clustering multi-hop routing protocol (PUCMR) (Hou et al., 2017). However, these protocols rely on a clustering approach, which may not be suitable for linear OGI pipeline monitoring scenarios. Linear IoT networks have their own challenges in terms of limited communication routes. Some pipelines utilize satellite communication which offers high latency, while others employ specialized long-range wireless devices that might not always be feasible. Moreover, smart devices have energy constraints, demands energy-efficient, reliable, and low-latency data transmission mechanisms.

Sarr et al. (2017) introduced a robust discovery, addressing, and routing protocol designed for dynamic linear wireless sensor networks (WSN). It offers the ability to add new spare nodes or branches of nodes. Sink nodes (or entire branches) can be incorporated either during the initialization of the linear network or at a later stage, using Dynamic DiscoProto, to address network overload or latency issues. It employs a hop count metric, facilitates traffic load balancing towards the sinks, reduces data packet latency, and enhances data packet delivery rates. However, it may introduce latency in dense WSNs with numerous overlapping branches. The proposed algorithm supports reallocation of new address blocks, ensuring efficient data routing throughout the network. Congestion of linear networks is still a challenge in this protocol. With the exponential growth of real-time data, many oil operators struggle to extract useful insights from the vast amounts of sensor values (which may be repeated) that must be routed to the cloud or control center, causing network congestion (Al-Sabaeei et al., 2023; Haseeb-Ur-Rehman et al., 2021). Hence, a data optimization mechanism is also necessary before sending data for analysis or action.

Recently, Bomgni et al. (2023) introduced a protocol, known as ESPINA, which puts emphasis on ensuring a network with low energy consumption, linear computational cost, and heightened security. However, it is designed for the random deployment of IoT networks, where nodes typically have multiple communication routes available. In contrast, linear deployment poses its own set of challenges due to limited communication routes, necessitating a focus on reliable communication. Furthermore, in the context of pipeline monitoring, low-latency communication for critical data becomes imperative. ESPINA, while excelling in energy efficiency and security for random deployments, does not address the crucial aspects of reliability and low-latency communication.

MultiHopFast is an another algorithm presented by Mtopi et al. (2023). This algorithm strategically alleviates the computing burden on IoT nodes through smart load balancing, ensuring scalability in IoT networks. However, it is tailored for cluster-based IoT deployments, addressing challenges related to network load, scalability, and time efficiency. Similarly, a protocol introduced by Farooq (2020) enhances the energy efficiency of a network but is specifically designed for cluster-based network, overlooking considerations of reliability and low-latency communication.

Additionally, Ali et al. (2023) introduced a Fog node in an IoT healthcare infrastructure, with a primary focus on power consumption as a key determinant. The authors propose a mathematical formulation to optimize the deployment of two heterogeneous gateways in healthcare infrastructure, aiming to minimize transmission power and infrastructure costs. However, it is crucial to note that this optimization target differs significantly from the reliability and low latency communication requirements of OGI pipeline monitoring through IoTs (Ali et al., 2023).

IoT solutions have emerged as a cost-effective and efficient method for data gathering. However, OGI pipelines often span vast distances, including remote and communication-limited areas like deserts where IoTs are particularly vulnerable to end-to-end delays and congestion because they possess a limited number of paths leading to the sink. This fact drastically changes the problem definition of such IoT network in terms of energy efficiency, fault tolerance, and low latency (Shuja et al., 2021). Table 1 clearly indicates that existing protocols address various parameters, with energy efficiency being a common focus. However, none of these protocols addresses the collective aspects of energy efficiency, reliability, and low-latency communication for linear networks considering the challenges of OGI pipeline monitoring scenarios.

Table 1 Packet type & its characteristics.

Ref	Contributions	Challenges	
		Energy efficiency	Fault tolerance	Low latency	MM/EM	Simulations	
Aba et al. (2021)	Dedicated wireless communication device, (Assumption)Internet availability along OGI pipeline, cloud-based decisions, Update every two minutes, Signal gathered every 15 seconds (sleep mode)	✓	×	×	×	×	
Spandonidis et al. (2022)	Energy-efficient WSN for OGI pipelines, Two distinct AI models are used to detect leakage. Sensor communication along long pipelines in diverse environments is a challenge	×	×	×	×	✓	
Singh et al. (2021)	2.4 GHz-based Zigbee and 433 MHzbased LoRa, Zigbee is a low-power wireless communication technology based on IEEE 802.15.4 (PAN), No energy consumption comparisons, Fault tolerance mechanism is implemented	✓	✓	×	×	×	
Khan et al. (2017)	Smart objects are used, radio transceiver for short-range communication, no energy-efficient mechanism, Reliable and low latency Communication mechanism is proposed but no implementation	×	✓	✓	×	×	
Ayeni & Ayogu (2020)	Four-tier IoT-based architecture, No current methods have been compared to the design, No actual simulations/implementations,No fault-tolerant, low latency, or energy-efficient IoT communication mechanism	×	×	×	×	×	
Chen, Zhao & Chen (2010)	-Clustering Approach	✓	×	×	×	×	
Hou et al. (2017)	-Clustering Approach	✓	×	×	×	×	
Sarr et al. (2017)	Protocol for linear WSN, focuses on scalibility, latency is still a challenge for dense WSN, creates congestion in the network	✓	×	×	×	×	
Bomgni et al. (2023)	focuses on energy efficiency, computational cost, and security, not for linear deployments	✓	×	×	×	×	
Mtopi et al. (2023)	Proposes algorithm for smart load balancing and scalability	✓	×	×	×	×	
Farooq (2020)	Not for linear deployments	✓	×	×	×	×	
Ali et al. (2023)	Introduces a Fog node in an IoT healthcare infrastructure, aiming to minimize transmission power and infrastructure costs	✓	×	×	✓	×	

IoTs system design for OGI pipelines

Multiple IoT devices are placed in a linearly on OGI pipelines. These IoT devices are typically powered by batteries and have limited energy resources. Since the pipelines traverse hazardous environments where recharging or replacing batteries is difficult, it becomes crucial to ensure energy-efficient communication in order to prolong the network’s lifespan. For this reason in our proposed system, we categorize IoT nodes into two types: simple (SN) and master nodes (MN).

Sensing nodes and master nodes characteristics

SNs are responsible for sensing data and transmitting it to MN. However, MN can do both sensing and gathering of data from its nodes and then transmit it to the upper layer. To make a low-cost and energy-efficient network, simple SNs are enabled with short-range communication whereas, the range of MNs is made adaptive to communicate with UAVs. Additionally, MNs are deployed at a hop distance of ten nodes from each other. Therefore, the number of SNs is ten times of MNs in the network (Mahbub & Shubair, 2023).

Functional architecture of IoT nodes

An IoT sensing node in comprised of the following units: sensing, location finding unit, processing unit, transceiver, battery, and memory, temperature sensor, pressure sensor, ultrasonic sensor for corrosion and erosion, and flow sensor: special piezo sensors for highly accurate flow measurements are embedded in the sensing unit of the IoT node.

Energy consumption model

IoT nodes use energy while transmitting and receiving messages. To analyze the different data delivery models, we must define the energy needed for transmitting and receiving messages in the IoTs network. For this purpose, we use the same radio-energy dissipation model as the one used by Gupta & Shekokar (2016) and Anand et al. (2016). Energy and distance between the source and destination node are directly proportional to each other. The smaller the distance, the lesser the energy consumed, and vice versa. However, there is a defined threshold distance between transmitting and receiving nodes given by Eq. (1): (1) do=Efs/Emp

where Efs is the free space energy and Emp is the multipath energy. If node A wants to forward a message to node B and their distance is more than this threshold distance do, then the energy required to send or receive a data packet is given by Eq. (2). (2) Econ=PL×Etx+Emp×PL×dAB4

If the distance between nodes is less than the specified threshold, the communication is happening in the near-field and the energy consumed can be calculated by Eq. (3). (3) Econ=PL×Etx+Emp×PL×dAB2

where Econ is the energy consumed, PL is the Packet Length, Etx is the transmission energy, dAB defines the distance between nodes A and B.

IoTs system working model

The detailed functioning model of IoTs is presented in this section. Based on the importance of the message, this layer has several objectives to achieve.

• The primary focus revolves around optimizing energy efficiency while transmitting regular OGI parameter values to ensure the network’s long-term sustainability.

• The paramount goal shifts to achieving low latency for critical data values.

• Fault tolerance and reliability with minimal data loss is a significant concern in both aforementioned scenarios.

To achieve all these objectives, we first design a basic mechanism for network deployment and route discovery, complemented by modules dedicated to enhancing reliability and fault tolerance. In conclusion, we present a suite of corresponding algorithms that encapsulate and facilitate these diverse endeavors.

IoTs network deployment

In general, pump stations are built every 20 to 100 miles along a pipeline, based on the terrain, the pipeline and station’s capacity, and the type of commodity being transported. The recommended distance between the sensor and sink nodes is [0–200(m)]. Therefore, between two consecutive pump stations, several master nodes (IoTs) are deployed. We consider 100 m length between two master nodes (Zhang et al., 2014). Every IoT node possesses a constrained sensing and communication radius. Let Rs denote the sensing range of IoT modules and Rc represent the communication range. In deploying IoT network, meeting the coverage requirements is an important aspect. The neighboring nodes should be deployed within the communication range (Rc) of a specific node. Therefore, according to Paulswamy & Kaluvan (2020), IoT modules must be deployed linearly at a distance of 3×Rs on pipelines. The communication range Rc must be greater than the distance of the nodes i.e., 3×Rs. Since it requires several sensor nodes between neighboring nodes to achieve connectivity, the geometry Rs<Rc≤3×Rs coverage approach requires a greater number of sensor nodes to cover the entire region. However, the Rc>3×Rs coverage method requires fewer sensors to cover the entire area because it does not need as many sensors to establish connectivity between neighbors.

Link establishment

After the network deployment phase, the link establishment phase is initiated by the master nodes (MN). The IoT nodes maintain the node-ID (IDn), master Node-ID (mID)), hop-count (hc), and next-hop (nh) address while creating route towards MN. The network route construction process is carried out by the nodes that receive this HELLO message by sending more HELLO messages. The ith Master node broadcasts Hello message to its nearest node. Hello message contains the ID of sending node (IDn), Master node ID (mID)), and hop-count (hc). Initially, the hop count is set to 0 (zero). As the message progresses through each subsequent node, the hop count is incremented by 1. Given our linear network configuration and the assumption that the communication range of sensors corresponds to their immediate next hop, the transmission of the HELLO message is confined to the immediate next-hop node. Upon receipt of the Hello message, the neighboring node resets its corresponding IDn and mID, increments hc by 1, updates nh towards MN, and continues this sequence until the Hello message reaches the other master node (MNj). Similarly, (MNj) establishes the second route by initiating its own HELLO message.

Each node knows its available energy, position, and its two routes towards master nodes 1 and 2 along with nearby nodes in its radio range after the link establishment process. Unlike other structure-based topology constructions, link establishment is only created once at the beginning and does not need to be repeated. We have proposed mechanisms to inform the node if it is about to die in advance. Therefore, the proposed approach will manage topological changes without reconstructing the topology. This will save a significant amount of energy. The operational details of this mechanism are delineated in Algorithm 1 .

Data transmission

IoT devices deployed along pipelines are tasked with monitoring temperature, pressure, and flow rate values. These parameters adhere to predefined minimum and maximum ranges, as previously elaborated in the introduction. In order to optimize energy consumption, IoT devices periodically gather sensor data. Specifically, temperature, pressure, and flow rate data are collected at intervals of 2 s, while ultrasonic data is acquired every 5 min.

Packet type: Let [Ti, Tf] and [Pi, Pf] be the normal temperature and pressure ranges. Tt and Pt be the sensed temperature and pressure at time t. Ω is used to denote the importance of a message. where Ω ∈ (0, 1). If Tt ∈ [Ti, Tf] and Pt ∈ [Pi, Pf], then Ω = 0. This implies that the message is classified as normal, and the value of Ω is assigned to 0. Conversely, if Tt⁄ ∈ [Ti, Tf] or Pt⁄ ∈ [Pi, Pf], then Ω = 1.

If either Tt or Pt falls outside the designated range, it signifies an urgent message, prompting the assignment of Ω to 1. The attributes of packet types are outlined in Table 2. The determination of Ω values is carried out by the originating node. Considering message significance, we introduce two data dissemination models. Notably, the reliability module is seamlessly integrated within both routing strategies.

Table 2 Packet type & its characteristics.

Sr#	Packet type	Attributes	Ranges	Constraints	Value	
1.	Normal	-Temperature -Pressure -Flow rate	-[Ti, Tf] -[Pi, Pf]	- Tt∈ [Ti, Tf] - Pt∈ [Pi, Pf]	Ω = 0	
2.	Critical	-Temperature -Pressure -Flow rate	-[Ti, Tf] -[Pi, Pf]	- Tt∉ in [Ti, Tf] - Pt∉ in [Pi, Pf]	Ω = 1	

Important message-based routing protocol (IMRP) for (Ω = 0)

Let us consider a network with N sensing nodes, where N ∈ {1, 2, 3, …, N}, and M master nodes, where M ∈ {1, 2, 3, …, M}. It’s important to note that the number of master nodes (M) is always less than the total number of sensing nodes (N), specifically M < N. When Ω = 0, data transmission occurs at uniform time intervals. The entire time duration is segmented into discrete time slots denoted by t, where t ∈ T and T ∈ {0, 5, 10, 15, 20, …}. In instances where a sensing node (SN) is unable to complete transmission within the allotted time, data loss might occur due to buffer overflow, as sensing nodes are constrained by limited buffer capacity.

Step 1: After network deployment and link establishment phase, individual nodes acquire their respective locations and dual routes. These routes encompass a path to master node 1 and another to master node 2, each accompanied by a dedicated route table containing details of hc and nh.

Step 2: Let’s consider the scenario where the ith sensing node aims to transmit mi bits of sensory data with Ω = 0 to the Master Node within a time frame of T seconds. This endeavor entails the following scenarios:

Scenario 1: both neighbors operational: In this scenario, both routes of Node i possess sufficient energy reserves for data transmission. In such instances, Node i evaluates both routes and opts for the one with a lower hc value leading to MN. To enhance network reliability and energy efficiency, a mechanism is in place to detect nodes approaching energy depletion, denoted by energy levels falling below a predefined threshold γ. In such cases, the node enters a sleep mode, briefly reactivating solely for transmission within a limited timeframe. Additionally, it communicates with its neighboring nodes, indicating its availability for a duration of tx, while simultaneously notifying the master node of its energy status for necessary interventions. The selection of the next hop for Scenario #1 is visually depicted in Fig. 1.

Figure 1 Scenario 1: both neighbors operational for Ω = 0.

Scenario 2: single available route: In this scenario, if the preferred route is inaccessible due to energy depletion of the next-hop node (nh), Node i opts for route 2 for data transmission. To optimize energy consumption during transmission, Node i disseminates information to all nodes along route 2, notifying them of route 1’s unavailability. Subsequently, any node wishing to transmit data will directly opt for route 2, and vice versa. The operational mechanism is visually depicted in Fig. 2.

Figure 2 Scenario 2: single available route for Ω = 0.

____________________________ Algorithm 1 Important Message-based Routing Protocol (IMRP)_________________________________________  1:  t ← 0  2:  while t ⁄= 0 do  3:       while alive_nodes ⁄= 0 do  4:           Find alive_nodes  5:           packets_transmitted ← packets_transmitted + 1  6:           if En ≤ 0.05 then  7:                Add node to dead node  8:           end if  9:           DATA(a,b,c) Determine datatype 10:           if Ω = 0 then 11:                if mod(t,10)==0 then 12:                     if route1 is available then           ROUTE1(nh)            Add (pl ∗ (Etx)) to (Emp ∗ pl ∗ nnd(SN,NN)2)            Subtract it from En 13:                     else if route2 is available then           ROUTE2(nh)            Add (pl ∗ (Etx)) to (Emp ∗ pl ∗ nnd(SN,NN)2)            Subtract it from En 14:                     else           Buffer data            Wait for UAV and transmit            Add (pl ∗ (Etx)) to (Emp ∗ pl ∗ nnd(SN,NN)4)            Subtract it from En 15:                     end if 16:                end if 17:           else 18:                if route1 is available then           ROUTE1(nh)            Add (pl ∗ (Etx)) to (Emp ∗ pl ∗ nnd(SN,NN)2)            Subtract it from En 19:                else           Increase Rc 20:                     if nodeb/wSN&MNisalive then           Transmit data to that node            Add (pl ∗ (Etx)) to (Emp ∗ pl ∗ nnd(SN,NN)4)            Subtract it from En 21:                     else           Transmit data to MN            Add (pl ∗ (Etx)) to (Emp ∗ pl ∗ nnd(SN,NN)4)            Subtract it from En 22:                     end if 23:                end if 24:           end if 25:       end while 26:       t ← t + 5 27:  end while________________________________________________________________________________________________________

Scenario 3: both routes unavailable: When both routes are unavailable for data transmission, Node i enters a listening mode and stores data in its buffer for a span of n readings. Subsequently, it awaits the arrival of upper-layer UAVs within range, extending its communication range as necessary. This facilitates data exchange and the dissemination of information regarding the inoperable neighboring nodes for necessary actions as shown in Fig. 3.

Figure 3 Scenario 3: both routes unavailable for Ω = 0.

Figure 4 Scenario 2: single available route for Ω = 1.

Important message-based routing protocol (IMRP) for (Ω = 1)

In situations where the sensed packet type is deemed critical, meaning it falls outside the predefined normal temperature and pressure ranges, low latency data transmission becomes imperative. This urgency is aimed at mitigating potential risks to both lives and assets. To address such exigencies, we outline a series of scenarios employing distinct mechanisms:

Scenario #1: Both neighbors operational: It is the same as in the previous routing mechanism described for Ω = 0. If both routes are alive, the route with less hop count is considered for critical packet transmission to reach the master node as soon as possible. If the node has the remaining energy equivalent to threshold energy γ, it continues its transmission without delaying toward the nearest master node. Moreover, it notifies the master node about the energy status of the sending node.

Scenario #2: Single available route: if the route with less hop count is not available i.e., the route to the nearest master node is unavailable. In that case, the node will increase its transmission range and send data to the next-to-next-hop neighbor on the preferred route. In case no node between the nearest master node and the sending node is available, the node will send that critical data directly to the master node. Next hop selection mechanism for scenario 2, Ω = 1 is illustrated in Fig. 4.

Scenario #3: Both routes unavailable: For critical messages, the prime objective is to send data to the master node in minimum time to reduce loss. If both the neighbors are dead for transmission, the node will increase its range and transmit the data to an alive node between the master node and the sending node. In case all nodes in between are dead, sending node will directly transmit data to the master node or if a UAV from the upper layer is in range, it will transmit a message through the UAV as shown in Fig. 5. The Algorithm 1 describes the functionality of IMRP

Figure 5 Scenario 3: both routes unavailable for Ω = 1.

_________________________________________________________________________________________________________________________ Algorithm 2 Packet Optimization for Important Message based Routing Protocol (PO-IMRP)___  1:  t ← 0  2:  while t ⁄= 0 do  3:       while alive_nodes ⁄= 0 do  4:           Find alive_nodes  5:           packets_transmitted ← packets_transmitted + 1  6:           if En ≤ 0.05 then  7:                Add node to dead node  8:           end if  9:           DATA(a,b,c) Determine datatype 10:           if Ω = 0 then 11:                if mod(t,10)==0 then 12:                     if route1 is available then           ROUTE1(nh)            Add (pl ∗ (Etx)) to (Emp ∗ pl ∗ nnd(SN,NN)2)            Subtract it from En            Attach data part to packet and forward 13:                     else if route2 is available then           ROUTE2(nh)            Add (pl ∗ (Etx)) to (Emp ∗ pl ∗ nnd(SN,NN)2)            Subtract it from En            Attach data part to packet and forward 14:                     else           Buffer data            Wait for UAV and transmit            Add (pl ∗ (Etx)) to (Emp ∗ pl ∗ nnd(SN,NN)4)            Subtract it from En 15:                     end if 16:                end if 17:           else 18:                if route1 is available then           ROUTE1(nh)            Add (pl ∗ (Etx)) to (Emp ∗ pl ∗ nnd(SN,NN)2)            Subtract it from En            Attach data part to packet and forward 19:                else           Increase Rc 20:                     if nodeb/wSN&MNisalive then           Transmit data to that node            Add (pl ∗ (Etx)) to (Emp ∗ pl ∗ nnd(SN,NN)4)            Subtract it from En            Attach data part to packet and forward 21:                     else           Transmit data to MN            Add (pl ∗ (Etx)) to (Emp ∗ pl ∗ nnd(SN,NN)4)            Subtract it from En 22:                     end if 23:                end if 24:           end if 25:       end while 26:       t ← t + 5 27:  end while________________________________________________________________________________________________________

Packet optimization for important message-based routing protocol (PO-IMRP)

Data forwarding in industrial IoTs whose nodes are powered by batteries is significantly hampered by energy consumption. These nodes expend a lot of energy forwarding, receiving, and temporarily storing messages. In OGI, the critical messages must reach master nodes for quick actions. We have proposed Packet Optimization for an Important Message Routing Protocol (PO-IMRP) which significantly reduces energy consumption and increases the lifetime of the network by optimizing data packets.

Sensor nodes deployed on pipelines are constantly sensing and reporting readings to master nodes at equal time intervals. In PO-IMRP, once a particular node has some parameter readings to transmit, it selects the route and collects the data from all the nodes residing on the route. IoT node packets have two parts: header and data part. The header contains the info of IDn, MN, hc, nh, and route status. The data part contains the values of temperature, pressure, and flow rate. However, all the sensed data is heading toward master nodes. Neighbor nodes can combine their data with the packets for further transmission.

Let node A intend to transmit packet pid1 to MN, it selects the route with a lower hc, represented by node B. Once the packet reaches node B, which holds its own data packet pid2 for transmission, an optimization is employed. Rather than transmitting both packets individually, each with distinct headers, node B consolidates the data and prepares a new packet featuring a unified header. Given the shared destination for both packets, this approach significantly conserves energy that would have been expended in transmitting two separate packets with distinct headers. The scenario where nodes A and B possess distinct packets for transmission is depicted in Fig. 6. An illustrative example showcasing the merging of a data packet with an identical header at the subsequent hop is presented in Fig. 7. The operational procedure is outlined in Algorithm 2 .

Figure 6 PO-IMRP node A and B have two different packets.

Figure 7 PO-IMRP combined data packets at B.

PO-IMRP packet format

The optimized packet format and its size are illustrated in Fig. 8. The whole packet has two parts: Header and Data. The header part is comprised of five fields i.e., node ID, master node ID, hop-count, next-hop, and route status. Each field contains 1 byte and a total of 5 bytes. The Data part combines the data with one separator field. The first three fields are for the first data packet comprised of 6 bytes then a separator field with 1 byte and then again data fields (6 bytes). The packet size increases on each hop.

Figure 8 PO-IMRP packet format.

PO-IMRP time complexity

PO-IMRP must determine the criticality of the messages that need to be transmitted. Let N be the total number of nodes in the network and the type of message at any time t is 0 or 1. Therefore, the time complexity between two communicating nodes is O(1). For N node, it will be O(1N) and for the whole IoT network, it will be O(N2). In this research, we aim to increase a specific level of time complexity while decreasing the transmission delay of critical messages. We must do a query on message k to determine whether the message type is normal or critical. The time required to query the type is O(1). The temporal complexity of computing the revenue of a node forwarding every message in the node buffer is O if there are n messages in the buffer (nk). According to the volume of revenue generated by the node forwarding messages, the node decides the order of message forwarding. The merge sorting method used in the sorting process has an O time complexity (n log n). Normally, message sending mechanism time overhead is primarily utilized to traverse route, and its time complexity is O (n).

Simulations & Results

In this research, we use MATLAB for performing simulations and comparative analyses of proposed IMRP and PO-IMRP. Energy and number of transmissions comparison with LEACH protocol is performed to facilitate a straightforward evaluation and understanding of the novel protocol’s strengths and weaknesses in relation to a known and accepted baseline. We have implemented LEACH for linear networks for comparison.

Simulations setup

The setup for simulations consists of simple sensing nodes (SN) and master nodes (MN). Simple sensing nodes are the units that contain temperature, pressure, and flow rate sensors along with the transceivers. These units have some predefined sensing and communication ranges respectively. However, master nodes have the same characteristics as simple sensing nodes but with an adaptive communication range. In our architectural framework, SN transmits data to MN, which serve as sinks. Typically, the distance from sensor nodes to the sink should fall within the range of 0 to 200 m. Consequently, we position MNs at a distance of 100 m. The simulation field we use to represent the OGI pipeline is 100 × 5 meters with MNs at both ends. The sensing range Rs of IoT nodes is set to 4.5 m and the communication range Rc to 10 m. Given our previous discussion that SNs are deployed at a distance of 3∗Rs, each SN is positioned 8 m apart. SN sense data at uniform time intervals i.e., t is set to 5s. while the total time is segmented into discrete time slots i.e., T is set to 10s. The initial energy Einit for each IoT module is set to be 0.5 J. Data transmission and reception energies Et and Er is 10−6 J. Free-space Energy Efs is set to 10−11 J, multipath Energy Emp as 1.3∗10−15 J, and data aggregation Energy Eaggr as 5∗10−9 J. The parameters used in simulations are outlined in Table 3. Node deployment is visually illustrated in Fig. 9.

Table 3 Simulations parameters.

Sr#	Parameter	Value	
1.	OGI Field Size	100 * 5 m	
2.	MN-MN distance	100 m	
3.	SN-SN distance	8 m	
4.	R s	4.5 m	
5.	SN − Rc	10 m	
6.	MN- Rc	Adaptive	
7.	t	5 s	
8.	T	10 s	
9.	E init	0.5 J	
10.	E t	10−6 J	
11.	E r	10−6 J	
12.	E fs	10−12 J	
13.	E mp	1.3∗10−15 J	
14.	E aggr	5∗10−9 J	

Figure 9 Network deployment.

In our study, the sensing nodes SNs acquire parameter values at uniform time intervals, denoted as t seconds. When these sensed values fall within the predefined ranges for temperature, pressure, and flow rate, data transmission adheres to the discrete-time slot T. In cases where the sensed values deviate from the specified ranges, triggering the identification of critical data, immediate transmission ensues, bypassing the designated time slots. Consequently, within the context of monitoring OGI pipelines, two distinct data categories emerge: normal and critical. To cater to the importance of these data types, we have introduced two routing mechanisms, namely IMRP and PO-IMRP. Within this section, we carry out an energy comparison of these proposed techniques.

Energy consumption vs packets transmitted

In this section, we will perform a comparative analysis of the energy consumption associated with both proposed routing protocols, IMRP and PO-IMRP. Et signifies the energy utilized for packet transmission, while Er represents the energy used during packet reception.

For IMRP

At a cold start, all the nodes have sufficient energy for data transmission. In IMRP, each node transmits a single packet at a time with a fixed packet length (PL). The packet size for each node’s message is 5 bytes for the header and 6 bytes for data with a total packet size of 11 bytes.

The first half of the nodes transmit data to the first MN, while the remaining half directs their data toward the second MN. let’s calculate energy consumption for each of these segments independently.

First half energy consumption-IMRP: From the energy consumption Eqs. (2) and (3) discussed in the previous section: Packet length is directly proportional to energy consumed by an IoT node. (4) Econ∝PL

However, PL is constant in IMRP with a value equal to 11 bytes. Hence, the energy consumption rule for any ith node will be derived as: (5) Ei=i⋅Et+Er⋅PLwherei∈1,6

For, Node 1:E1=Et+Er⋅PL

Node 2:E2=2⋅Et+Er⋅PL

Node 3:E3=3⋅Et+Er⋅PL……

Node 6:E6=6⋅Et+Er⋅PL

Therefore, the total energy consumption for the first half is: Efirst-half= ∑i=16EiEfirst-half=E1+E2+E3+…+E6

Efirst-half = (Et + Er + 2Et + 2Er + 3Et + 3Er + 4Et + 4Er + 5Et + 5Er + 6Et + 6Er)⋅PL

Efirst-half = (21Et + 21Er)⋅PL

As Et = Er = E, (6) ∴Efirst-half=42E⋅PL

Second half energy consumption-IMRP: Similar to the first half, the second half of IoT nodes’ energy consumption will be the same, i.e., (7) Esecond−half=42E⋅PL

Total energy consumption-IMRP: The total Energy consumption can be calculated by combining Eqs. (6) & (7).

Etotal = Efirst-half + Esecond-half

Etotal = (42E⋅PL) + (42E⋅PL) (8) Etotal=84E⋅PL

Packets transmitted (IMRP): Each IoT node exactly transmits one packet i.e., pi = i. Therefore, the total number of packets transmitted in a single round is: Ptotal= ∑i=112piPtotal=p1+p2+p3+…+p12

∴Ptotal=12.

Node 1 transmits one packet at a time, each with a fixed packet length (PL). Likewise, Node 2 also engages in transmission, and the process is reciprocated for all nodes. The packet size for the message from each node comprises 5 bytes for the header and 6 bytes for data, resulting in a total packet size of 11 bytes. The hop-by-hop packet transmission mechanism for each node is illustrated in Table 4.

Table 4 IMRP packets transmission.

PID	Packet originating node	PL	nh	PL	nh	PL	nh	PL	nh	PL	nh	PL	nh	No. of sensed values sent	
1	1	11	MN											1	
2	2	11	1	11	MN									1	
3	3	11	2	11	1	11	MN							1	
4	4	11	3	11	2	11	1	11	MN					1	
5	5	11	4	11	3	11	2	11	1	11	MN			1	
6	6	11	5	11	4	11	3	11	2	11	1	11	MN	1	
7	7	11	8	11	9	11	10	11	11	11	12	11	MN	1	
8	8	11	9	11	10	11	11	11	12	11	MN			1	
9	9	11	10	11	11	11	12	11	MN					1	
10	10	11	11	11	12	11	MN							1	
11	11	11	12	11	MN									1	
12	12	11	MN											1	
Total	12	

For PO-IMRP

In the PO-IMRP protocol, the packet size is adaptive and not fixed. When a node initiates transmission, it gathers and appends the data portions of all the other nodes through which it hops toward the master node (MN). Initially, the packet length is 11 bytes, consisting of 5 bytes for the header and 6 bytes for the data segment. However, with each hop from one node to another, the packet length grows by 7 bytes, encompassing 6 data bytes from the current node’s data and an additional 1-byte separator. Therefore, initially,

PL = 11 bytes

With each successive hop, the packet length (PL) undergoes an update of 7 + PL. (9) PL=7+PL.

Let’s calculate the total energy consumption for a single round for PO-IMRP.

First half energy consumption PO-IMRP: The energy consumption rule for ith node will be:

Ei=iEt+iEr⋅7j+PL

where, i ∈ [1, 6] and j ∈ [0, 5]

For,

Node 1: i = 1j =0 ⟹E1=1Et+1Er⋅70+PL

Hence, (10) E1=Et+Er⋅PL

Put Et = Er = E and PL = pl1 in Eq. (10) (11) E1=2E⋅pl1

Similarly, (12) E2=2Et+2Er⋅7+PL

Put Et = Er = E and (7 + PL) = pl2 in Eq. (12) (13) E2=4E⋅pl2

Hence, Node 3:E3=3Et+3Er⋅14+PL⟹E3=6E⋅pl3……

Node 6:E6=6Et+6Er⋅35+PL⟹E6=12E⋅pl6

Therefore, the total energy consumption for the first half is: Efirst-half= ∑i=16EiEfirst-half=E1+E2+E3+…+E6

Efirst−half =  [2E⋅pl1] + [4E⋅pl2] + [6E⋅pl3] + [8E⋅pl4] + [10E⋅pl5] + [12E⋅pl6] (14) Efirst−half=E2pl1+4pl2+6pl3+8pl4+10pl5+12pl6

Second half energy consumption PO-IMRP: Similar to the first half, the second half of IoT nodes’ energy consumption will be the same, i.e., (15) Esecond−half=E2pl1+4pl2+6pl3+8pl4+10pl5+12pl6

Total energy consumption PO-IMRP The total Energy consumption can be calculated as:

Etotal=∑i=12Ei

Adding Eqs. (14) and (15)

Etotal = [E(2pl1 + 4pl2 + 6pl3 + 8pl4 + 10pl5 + 12pl6)] + [E(2pl1 + 4pl2 + 6pl3 + 8pl4 + 10pl5 + 12pl6)] (16) Etotal=Epl1+2pl2+3pl3+4pl2+5pl5+6pl6

Packets transmitted-PO-IMRP: In the PO-IMRP protocol, the packet length is not a fixed value; instead, it adapts to optimize packet transmission. When an IoT node initiates data transmission, it appends the data segment of each subsequent hopped IoT node, leading to an increased packet length at each hop until it reaches the destination. Consequently, the number of packets transmitted in a single round is higher compared to IMRP. To determine the total number of packets transmitted in a single round, consider the following: Node 1 initiates the process by transmitting a single packet with a length of 11 bytes to MN. As Node 2 transitions to the transmission phase, it dispatches an 11-byte packet to Node 1. Node 1 then appends additional sensed values to the packet before forwarding the extended packet to MN. Consequently, this process enables the transmission of two sets of data without causing network congestion i.e., an 11-byte packet by Node 2 and an 18-byte packet by Node 1 is transmitted to MN in a single transmission. This pattern repeats for the remaining nodes. A summarized representation of packets and transmissions for the entire round can be found in Table 5.

Table 5 PO-IMRP packets transmission.

P_ID	Packet originating node	PL	nh	PL	nh	PL	nh	PL	nh	PL	nh	PL	nh	No. of sensed values sent	
1	1	11	MN											1	
2	2	11	1	18	MN									2	
3	3	11	2	18	1	25	MN							3	
4	4	11	3	18	2	25	1	32	MN					4	
5	5	11	4	18	3	25	2	32	1	39	MN			5	
6	6	11	5	18	4	25	3	32	2	39	1	46	MN	6	
7	7	11	8	18	9	25	10	32	11	39	12	46	MN	6	
8	8	11	9	18	10	25	11	32	12	39	MN			5	
9	9	11	10	18	11	25	12	32	MN					4	
10	10	11	11	18	12	25	MN							3	
11	11	11	12	18	MN									2	
12	12	11	MN											1	
Total Sensed values transmitted in 12 packets	42	

Table 6 IMRP vs PO-IMRP results.

Factors	IMRP	PO-IMRP	
No. of transmissions	11,194	17,784	
No. of rounds	942	426	
No. of packets	11,194	<5,000	

Packets congestion control management

Our proposed IoTs routing mechanism, PO-IMRP offers packet control management by optimizing packets. Table 6 presents the total number of transmissions, rounds, and packets delivered by IMRP and PO-IMRP. IMRP accomplished 942 rounds and facilitated the transmission of 11,194 packets throughout the network’s lifetime. In contrast, PO-IMRP generated 426 packets while efficiently transmitting 17,784 data values through optimized packets. Through a comprehensive analysis of these statistics, it can be concluded that PO-IMRP performs well by delivering more data in fewer rounds and generating fewer packets in the network.

Efficiency calculation

PO-IMRP is 37.06 % more efficient than IMRP in transmitting sensed values by reducing packet generation in the network throughout the network’s lifetime. (17) Efficiency=37.06%

Effect on transmissions per round (IMRP vs PO-IMRP)

Figure 10 shows the relationship between transmissions and rounds. The peak values of both show that PO-IMRP performs more transmissions in fewer rounds than IMRP.

Figure 10 Transmissions per round.

Effect on energy consumption

Figure 11 describes the pattern between net energy of the network per round throughout the network lifetime. Although the energy consumption of PO-IMRP is more than IMRP the transmission in the network increases significantly. The net energies of IMRP and PO-IMRP after performing transmissions of round 1 are as shown in the Table 7. Hence, the Energy gap is calculated as: (18) Egap=5.9962−5.9932=0.003J

Figure 11 Net energy of network per round.

Performance analysis of PO-IMRP vs LEACH

We have performed a performance analysis of our proposed protocol PO-IMRP with LEACH. Low Energy Adaptive Clustering Hierarchy (LEACH) is one of the popular WSN routing protocols. It is proposed by Heintzelman. The idea behind choosing LEACH for comparison is that it is a renowned WSN routing protocol, capable of enhancing the lifetime of the network, and provides fault tolerance Mohapatra & Rath (2019). Our analysis involved implementing LEACH within the context of our linear WSN setup and associated parameters, enabling a comprehensive comparison of outcomes.

Operational nodes per transmission (PO-IMRP vs LEACH)

The relationship between operating nodes per round is shown in Fig. 12. In PO-IMRP, the first node dies after 392 rounds and 16,464 packets have been transmitted. However, in LEACH, the first node dies between 600–700 rounds and less than 1,000 packets have been transmitted. This indicates that PO-IMRP shows more transmissions than LEACH.

Table 7 Remaining energies of nodes for a single round.

Node	Energy-IMRP (J)	Energy PO-IMRP (J)	
1	0.4995	0.4988	
2	0.4996	0.4991	
3	0.4996	0.4994	
4	0.4997	0.4996	
5	0.4998	0.4998	
6	0.4999	0.4999	
7	0.4999	0.4999	
8	0.4998	0.4998	
9	0.4997	0.4996	
10	0.4996	0.4994	
11	0.4996	0.4991	
12	0.4995	0.4988	
Total energy	5.9962	5.9932	

Figure 12 Operating nodes per round (PO-IMRP vs LEACH).

Energy consumption per transmission:

Fig. 13 depicts the correlation between Energy and transmissions of the IMRP, PO-IMRP, and LEACH protocol. LEACH shows energy consumption between 0.01 to 0.22 J. However, PO-IMRP shows an average energy consumption of 7.066 × 10−3 J. This analysis demonstrates that PO-IMRP is more energy efficient than LEACH.

Figure 13 Energy vs transmissions (PO-IMRP vs LEACH).

conclusion

IoT sensors have gained significant attention in industrial and manufacturing applications. This is due to their capability to enable interaction between computer networks and the physical environment through the sensing of physical parameters. This study introduces a novel protocol called Packet Optimization for Important Message-based Routing Protocol (PO-IMRP). This protocol is tailored to tackle challenges linked to energy efficiency, fault tolerance, end-to-end delay, and optimized data transmission in monitoring OGI pipelines. PO-IMRP prioritizes critical data and transmits it to the control center with the highest priority for rapid response, mitigating potential losses in OGI assets. Furthermore, the protocol optimizes data packets to ensure an energy-efficient and congestion-free network. Simulation results showcase PO-IMRP’s impressive 37.06% efficiency in transmitting sensed values, achieved by minimizing packet generation throughout the network’s lifespan. Comparative analysis with the linear LEACH protocol, considering network lifetime and energy consumption, demonstrates PO-IMRP’s average energy consumption of 7.066 × 10−3 J, in contrast to LEACH’s energy consumption ranging from 0.01 to 0.22 J. This indicates that PO-IMEERP outperforms linear LEACH in terms of the average energy consumption and total number of packets transmission. Beyond the scope of OGI, PO-IMRP is adaptable to various other applications characterized by linear deployment, necessitating energy efficiency, fault tolerance, and low latency to prevent losses. Applications such as infrastructure monitoring, including roads, bridges, and railway tracks. Additionally, it can be adapted for diverse security and surveillance applications, such as border security. In the future, we plan to enhance this study in two ways: (1) incorporate UAVs as communication relays for efficient and dynamic communication routes to enhance fault tolerance and low latency communication in the proposed PO-IMRP; (2) energy harvesting through UAV when the residual energy of IoTs is about to deplete for longer network lifetime.

Supplemental Information

Supplemental Information 1 Matlab code for simulations

Additional Information and Declarations

Competing Interests

Author Contributions

Data Availability

Junaid Shuja is an Academic Editor for PeerJ

Sana Nasim Karam conceived and designed the experiments, performed the experiments, analyzed the data, performed the computation work, prepared figures and/or tables, authored or reviewed drafts of the article, and approved the final draft.

Kashif Bilal conceived and designed the experiments, performed the experiments, performed the computation work, authored or reviewed drafts of the article, and approved the final draft.

Abdul Nasir Khan analyzed the data, authored or reviewed drafts of the article, and approved the final draft.

Junaid Shuja analyzed the data, prepared figures and/or tables, authored or reviewed drafts of the article, and approved the final draft.

Said Jadid Abdulkadir analyzed the data, prepared figures and/or tables, authored or reviewed drafts of the article, and approved the final draft.

The following information was supplied regarding data availability:

The code is available in the Supplemental File.

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
