# Peer review of "Energy-efficient routing protocol for reliable low-latency Internet of Things in oil and gas pipeline monitoring"

_PeerJ Computer Science, doi:10.7717/peerj-cs.1908_

## Round 0.1 · original submission · Major Revisions

Dear authors
Your manuscript needs a major revisions as pointed out by the experts, therefore please carefully revise the paper according to comments and resubmit.
Please also check the language correctness in revised version
What are the practical implications and values of your work.
Thanks

**Language Note:** The Academic Editor has identified that the English language must be improved. PeerJ can provide language editing services - please contact us at copyediting@peerj.com for pricing (be sure to provide your manuscript number and title). Alternatively, you should make your own arrangements to improve the language quality and provide details in your response letter. – PeerJ Staff

Reviewer 1 ·

Basic reporting

In this article, authors present an energy-efficient routing protocol for IoT Network. Proposed routing protocol is specifically designed for an IoT Network which offers reliability and low-latency and used for OGI pipeline monitoring. Overall writing style of the article is good.

Experimental design

Ok

Validity of the findings

Ok

Additional comments

I suggest followings suggestions should be addressed before publishing this article.
Below are some comments/suggestions regarding the article.
1) A proper number should be written with each section in order to avoid any confusion.
2) Figure/diagram/table/algorithm should appear on the top of page where it is first time referred in the text.
3) Title includes “Reliable and Low-Latency IoT Network”. How reliability and low latency is ensured by proposed technique?
4) Section “Related Work”, Table 1, rather than just referring to the table in middle of text. It should be properly discussed at the end of this section. Moreover, comparison table should have comparison of all the techniques discussed in this section. Also, broader scope should be included in the discussion about energy/power consumption such as but NOT limited to (i) ESPINA: efficient and secured protocol for emerging IoT network applications (ii) MultiHop optimal time complexity clustering for emerging IoT applications (iii) Power-Aware Fog Supported IoT Network for Healthcare Infrastructure Using Swarm Intelligence-Based Algorithms.
5) Section “Energy Consumption Model”, Is this your own energy model? If yes, also add the logic and rationale behind this model (provide mathematical modelling). If not, add the reference.
6) A paragraph should be attended after each Algorithm to discuss its main components.
7) Section, Simulation and Results, which Matlab library/module is used to simulate the given environment?
8) Section, Simulation and Results, Authors have compared their work with LEACH, which is a very old protocol. It is suggested that authors should compare their work with at least 2 or 3 latest state-of-the-art protocols. Moreover, rationale behind presented results is missing. How the proposed algorithm achieved so much significantly better results?

·

Basic reporting

In the paper titled “Energy-Efficient Routing Protocol for Reliable Low-latency IoT Network in OGI Pipeline Monitoring” authors propose a routing mechanism to cater energy efficiency, fault tolerance and low latency surveillance through IoTs in OGI pipelines. Overall, the article is well written. However, the technical writing of article needs major organization/presentation changes with the following points:

• Consider replacing the underline word with crucial in the following line of abstract:
“Therefore, maintaining an effective pipeline maintenance system is critical for ensuring a safe and sustainable energy supply.”
• Add comma after PO-IMRP in the following line of abstract: In PO-IMRP, nodes are aware of their energy status and communicate node’s depletion status timely for network robustness.
• Replace POIMRP with PO-IMRP’s in the line: Comparative analysis against linear LEACH highlights PO-IMRP’s superior performance in terms of total packet transmission.
• There is no discussion on research problems in the introduction. What is lacking in previous studies? What is the research gap. A paragraph needs to be added. Then what does the proposed solution offer? How does it solve the problem?
• Capitalize the first word of gas in the first line of Introduction: Oil and Gas Industry (OGI) pipelines are the primary resource for global energy transportation.
• Consider rewriting this line in introduction: There are currently 2.2 million miles of pipelines that are deployed to fulfill the continuous demand for OGI products around the World.
• Consider limiting the use of However in the second paragraph of Introduction.
• Consider rewriting: “They can be installed on transmission pipelines in challenging environments and from remote locations for round-the-clock monitoring and offer accurate pipeline leakage detection data.”
• Consider replacing the highlighted text with “thereby reducing traffic”: “Optimal data transmission is done by removing redundant data and by maintaining a time-lapse for normal data acquisition and transmission, hence also reducing the traffic in the main network.”
• In the section “Sensing Nodes and Master Nodes Characteristics”, correct the format of the following reference: “Mahbub and Shubair (2023)” .
• In the section “IoT System Working Model” consider writing the objectives into Numbered items to create an impact on the reader.
• There is no clear description for Table 4 and 5. Consider adding detailed transmission of packets for reader’s understanding.
• Capitalize the first word of the following where referred: equation, table, and figure in the whole article.

Experimental design

no comment

Validity of the findings

no comment

Reviewer 3 ·

Basic reporting

The paper, entitled "Energy-Efficient Routing Protocol for Reliable Low-latency IoT Network in OGI Pipeline Monitoring," introduces a routing mechanism designed to address the three crucial factors of IoT: energy efficiency, fault tolerance, and low-latency specific to Oil and Gas pipeline monitoring scenario.
The authors could improve the manuscript by correcting a couple of aspects:

Experimental design

1. Correct the references in the whole article: e.g.,
• Mahbub and Shubair (2023) at line#191
• Zhang et al. (2014) at line# 224
• Paulswamy and Kaluvan (2020) at line#232
• Mohapatra and Rath (2019) at line485
2. The abstract states that they proposed a priority based optimal data routing protocol. What is the significance of priority here? What is the context of Priority based routing?
3. The relationship between Rs (sensing range) and Rc (communication range) should be explained more cohesively.
4. The section on the "Energy Consumption Model" might require clearer explanations and context for the equations presented. Symbols are not adequately declared. This section needs careful revisions.
5. Consider adding specific recommendations in future work based on the study's outcomes.
6. The author should discuss the broader implications of the findings beyond the immediate scope.
7. Table 1 should use some symbol’s instead of Y and N.

Validity of the findings

8. Rephrase line#121
“The communication of damaged data acquired by sensors to the monitoring room is accomplished wirelessly, allowing real-time pipeline monitoring from anywhere in the world (around the World).”
9. Place , at line 459
“Our proposed IoTs routing mechanism, PO-IMRP offers packet control management by optimizing packets.”
10. Table 8 is referred to on page 17 and it is on page 20. Similarly Figure 14 is referred to on page 19 and is placed on page 21. Reorganize the placement for reader’s understanding.
11. I suggest adding comparative analysis details of proposed routing protocol PO-IMRP against linear LEACH in abstract. It will help the reader to grasp the understanding of article’s analysis.

---

## Round 0.2 · accepted · Accept

Dear authors,

Thanks for your resubmission of the updated article, I have now evaluated the responses and also the reviewers' recommendations.

Based on their input, I'm pleased to notify you about the acceptance of your article. Thank you for your fine contribution.

Reviewer 1 ·

Basic reporting

Good shape

Experimental design

Good shape

Validity of the findings

Good shape

Additional comments

My comments are addressed.

·

Basic reporting

The requested review was handled well, and I suggest that this paper be accepted.

Experimental design

No extra comments, the performance is well discussed

Validity of the findings

No extra comments.

Reviewer 3 ·

Basic reporting

However the text is improved, yet authors are suggested to considering giving it another read for improving text and grammar.

Experimental design

Results are clearly stated and the methodology is well defined.

Validity of the findings

Results are clearly stated and the methodology is well defined.

Additional comments

All the previous comments have been answered and the paper is in good shape for further consideration.